# VISUALIZING THE LOSS LANDSCAPE OF NEURAL NETS

## ABSTRACT

Neural network training relies on our ability to find "good" minimizers of highly non-convex loss functions. It is well known that certain network architecture designs (e.g., skip connections) produce loss functions that train easier, and well-chosen training parameters (batch size, learning rate, optimizer) produce minimizers that generalize better. However, the reasons for these differences, and their effect on the underlying loss landscape, is not well understood.

In this paper, we explore the structure of neural loss functions, and the effect of loss landscapes on generalization, using a range of visualization methods. First, we introduce a simple "filter normalization" method that helps us visualize loss function curvature, and make meaningful side-by-side comparisons between loss functions. Then, using a variety of visualizations, we explore how network architecture effects the loss landscape, and how training parameters affect the shape of minimizers.

## 1 INTRODUCTION

Training neural networks requires minimizing a high-dimensional non-convex loss function – a task that is hard in theory, but sometimes easy in practice. Despite the NP-hardness of training general neural loss functions (Blum & Rivest, 1989), simple gradient methods often find global minimizers (parameter configurations with zero or near-zero training loss), even when data and labels are randomized before training (Zhang et al., 2017). However, this good behavior is not universal; the trainability of neural nets is highly dependent on network architecture design choices, the choice of optimizer, variable initialization, and a variety of other considerations. Unfortunately, the effect of each of these choices on the structure of the underlying loss surface is unclear. Because of the

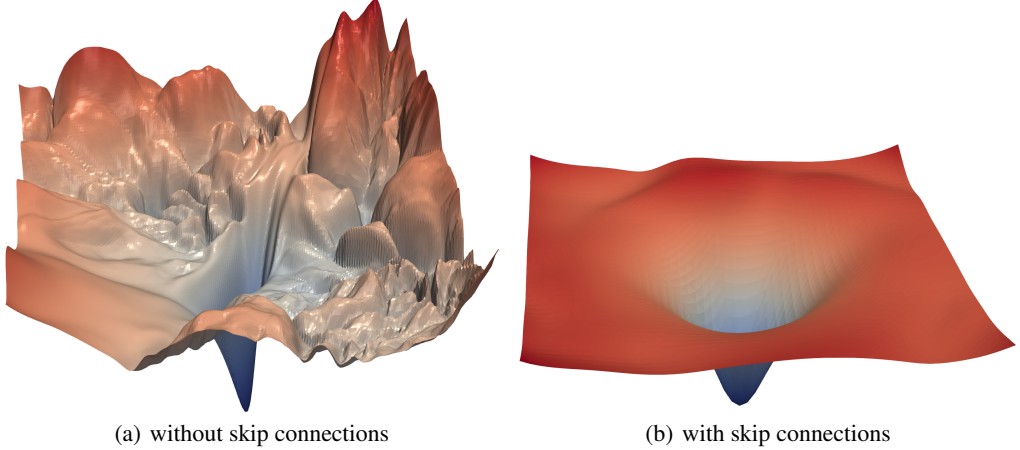

(a) without skip connections                    (b) with skip connections

Figure 1: The loss surfaces of ResNet-56 with/without skip connections. The vertical axis is logarithmic to show dynamic range. The proposed filter normalization scheme is used to enable comparisons of sharpness/flatness between the two figures.

prohibitive cost of loss function evaluations (which requires looping over all the data points in the training set), studies in this field have remained predominantly theoretical.

Our goal is to use high-resolution visualizations to provide an empirical characterization of neural loss functions, and to explore how different network architecture choices affect the loss landscape. Furthermore, we explore how the non-convex structure of neural loss functions relates to their trainability, and how the geometry of neural minimizers (i.e., their sharpness/flatness, and their surrounding landscape), affects their generalization properties.

To do this in a meaningful way, we propose a simple "filter normalization" scheme that enables us to do side-by-side comparisons of different minima found by different methods. We then use visualizations to explore sharpness/flatness of minimizers found by different methods, as well as the effect of network architecture choices (use of skip connections, number of filters, network depth) on the loss landscape. Out goal is to understand how differences in loss function geometry effect the generalization of neural nets.

## 1.1 CONTRIBUTIONS

In this article, we study methods for producing meaningful loss function visualizations. Then, using these visualization methods, we explore how loss landscape geometry effects generalization error and trainability. More specifically, we address the following issues:

- We reveal faults in a number of visualization methods for loss functions, and show that simple visualization strategies fail to accurately capture the local geometry (sharpness or flatness) of loss function minimizers.

- We present a simple visualization method based on "filter normalization" that enables side-by-side comparisons of different minimizers. The sharpness of minimizers correlates well with generalization error when this visualization is used, even when making sharpness comparisons across disparate network architectures and training methods.

- We observe that, when networks become sufficiently deep, neural loss landscapes suddenly transition from being nearly convex to being highly chaotic. This transition from convex to chaotic behavior, which seem to have been previously unnoticed, coincides with a dramatic drop in generalization error, and ultimately to a lack of trainability.

- We show that skip connections promote flat minimizers and prevent the transition to chaotic behavior, which helps explain why skip connections are necessary for training extremely deep networks.

- We study the visualization of SGD optimization trajectories. We explain the difficulties that arise when visualizing these trajectories, and show that optimization trajectories lie in an extremely low dimensional space. This low dimensionality can be explained by the presence of large nearly convex regions in the loss landscape, such as those observed in our 2-dimensional visualizations.

## 2 THEORETICAL BACKGROUND & RELATED WORK

Visualizations have the potential to help us answer several important questions about why neural networks work. In particular, why are we able to minimize highly non-convex neural loss functions? And why do the resulting minima generalize?

Because of the difficultly of visualizing loss functions, most studies of loss landscapes are largely theoretical in nature. A number of authors have studied our ability to minimize neural loss functions. Using random matrix theory and spin glass theory, several authors have shown that local minima are of low objective value (Dauphin et al., 2014; Choromanska et al., 2015). It can also be shown that local minima are global minima, provided one assumes linear neurons (Hardt & Ma, 2017), very wide layers (Nguyen & Hein, 2017), or full rank weight matrices (Yun et al., 2017). These assumptions have been relaxed by Kawaguchi (2016) and Lu & Kawaguchi (2017), although some assumptions (e.g., of the loss functions) are still required. Soudry & Hoffer (2017); Freeman & Bruna (2017); Xie et al. (2017) also analyzed shallow networks with one or two hidden layers under mild conditions.

Another approach is to show that we can expect good minimizers, not simply because of the endogenous properties of neural networks, but because of the optimizers. For restricted network classes such as those with one hidden layer, with some extra assumptions on the sample distribution, globally optimal or near-optimal solutions can be found by common optimization methods (Soltanolkotabi et al., 2017; Li & Yuan, 2017; Tian, 2017). For networks with specific structures, Safran & Shamir (2016) and Haeffele & Vidal (2017) show there likely exists a monotonically decreasing path from an initialization to a global minimum. Swirszcz et al. (2017) show counterexamples that achieve "bad" local minima for toy problems.

Also of interest is work on assessing the sharpness/flatness of local minima. Hochreiter & Schmidhuber (1997) defined "flatness" as the size of the connected region around the minimum where the training loss remains low. Keskar et al. (2017) propose $\epsilon$-sharpness, which looks at the maximum loss in a bounded neighborhood of a minimum. Flatness can also be defined using the local curvature of the loss function at a critical point. Keskar et al. (2017) suggests that this information is encoded in the eigenvalues of the Hessian. However, Dinh et al. (2017) show that these quantitative measure of sharpness are problematic because they are not invariant to symmetries in the network, and are thus not sufficient to determine its generalization ability. This issue was addressed in Chaudhari et al. (2017), who used local entropy as a measure of sharpness. This measure is invariant to the simple transformation used by Dinh et al. (2017), but difficult to quantify for large networks.

Theoretical results make some restrictive assumptions such as the independence of the input samples, or restrictions on non-linearities and loss functions. For this reason, visualizations play a key role in verifying the validity of theoretical assumptions, and understanding loss function behavior in real-world systems. In the next section, we briefly review methods that have been used for this purpose.

## 3    THE BASICS OF LOSS FUNCTION VISUALIZATION

Neural networks are trained on a corpus of feature vectors (e.g., images) $\{x_i\}$ and accompanying labels $\{y_i\}$ by minimizing a loss of the form

$$L(\theta) = \frac{1}{m} \sum_{i=1}^{m} \ell(x_i, y_i; \theta)$$

where $\theta$ denotes the parameters (weights) of the neural network, the function $\ell(x_i, y_i; \theta)$ measures how well the neural network with parameters $\theta$ predicts the label of a data sample, and $m$ is the number of data samples.

Neural nets contain many parameters, and so their loss functions live in a very high-dimensional space. Unfortunately, visualizations are only possible using low-dimensional 1D (line) or 2D (surface) plots. Several methods exist for closing this dimensionality gap.

**1-Dimensional Linear Interpolation**    One simple and lightweight way to plot loss functions is to choose two sets of parameters $\theta_1$ and $\theta_2$, and plot the values of the loss function along the line connecting these two points. We can parameterize this line by choosing a scalar parameter $\alpha$, and defining the weighted average

$$\theta_\alpha = (1 - \alpha)\theta_1 + \alpha\theta_2.$$

Finally, we plot the function $f(\alpha) = L(\theta_\alpha)$. This strategy was taken by Goodfellow et al. (2015), who studied the loss surface along the line between a (random) initial guess, and a nearby minimizer obtained by stochastic gradient descent. This method has been widely used to study the "sharpness" and "flatness" of different minima, and the dependence of sharpness on batch-size (Keskar et al., 2017; Dinh et al., 2017). Smith & Topin (2017) use the same 1D interpolation technique to show different minima and the "peaks" between them, while Im et al. (2016) plot the line between minima obtained via different optimizers.

The 1D linear interpolation method suffers from several weaknesses. First, it is difficult to visualize non-convexities using 1D plots. Indeed, the authors of (Goodfellow et al., 2015) found that loss functions appear to lack local minima along the minimization trajectory. We will see later, using 2D methods, that some loss functions have extreme non-convexities, and that these non-convexities correlate with the difference in generalization between different network architectures. Second, this

method does not consider batch normalization or invariance symmetries in the network. For this reason, the visual sharpness comparisons produced by 1D interpolation plots may be misleading; this issue will be explored in depth in Section 5.

**2D Contour Plots**    To use this approach, one chooses a center point $\theta^*$ in the graph, and chooses two direction vectors, $\delta$ and $\eta$. One then plots a function of the form $f(\alpha) = L(\theta^* + \alpha\delta)$ in the 1D (line) case, or

$$f(\alpha, \beta) = L(\theta^* + \alpha\delta + \beta\eta) \tag{1}$$

in the 2D (surface) case. This approach was used in (Goodfellow et al., 2015) to explore the trajectories of different minimization methods. It was also used in (Im et al., 2016) to show that different optimization algorithms find different local minima within the 2D projected space.

Because of the computational burden of 2D plotting, these methods generally result in low-resolution plots of small regions that have not captured the complex non-convexity of loss surfaces. Below, we use high-resolution visualizations over large slices of weight space to visualize how network design affects non-convex structure.

## 4    PROPOSED VISUALIZATION: FILTER-WISE NORMALIZATION

This study relies heavily on plots of the form (1) produced using random direction vectors, $\delta$ and $\eta$, each sampled from a random Gaussian distribution with appropriate scaling (described below).

While the "random directions" approach to plotting is simple, it cannot be used to compare the geometry of two different minimizers or two different networks. This is because of the *scale invariance* in network weights. When ReLU non-linearities are used, the network remains unchanged if we (for example) multiply the weights in one layer of a network by 10, and divide the next layer by 10. This invariance is even more prominent when batch normalization is used. In this case, the size (i.e., norm) of a filter is irrelevant because the output of each layer is re-scaled during batch normalization. For this reason, a network's behavior remains unchanged if we re-scale the weights.

Scale invariance prevents us from making meaningful comparisons between plots, unless special precautions are taken. A neural network with large weights may appear to have a smooth and slowly varying loss function; perturbing the weights by one unit will have very little effect on network performance if the weights live on a scale much larger than one. However, if the weights are much smaller than one, then that same one unit perturbation may have a catastrophic effect, making the loss function appear quite sensitive to weight perturbations. Keep in mind that neural nets are scale invariant; if the small-parameter and large-parameter networks in this example are equivalent (because one is simply a re-scaling of the other), then any apparent differences in the loss function are merely an artifact of scale invariance. This scale invariance was exploited by Dinh et al. (2017) to build pairs of equivalent networks that have different apparent sharpness.

To remove this scaling effect, we plot loss functions using filter-wise normalized directions. To obtain such directions for a network with parameters $\theta$, we begin by producing a random Gaussian direction vector $d$ with dimensions compatible with $\theta$. Then we normalize each filter in $d$ to have the same norm of the corresponding filter in $\theta$. In other words, we make the replacement

$$d_i \leftarrow \frac{d_i}{\|d_i\|} \|\theta_i\|,$$

where $d_i$ represents the $i$th filter of $d$ (not the $i$th weight), and $\|\theta_i\|$ denotes the Frobenius norm of the $i$th filter of $\theta$. Note that the filter-wise normalization is different from that of (Im et al., 2016), which normalize the direction without considering the norm of individual filters.

The proposed scaling is an important factor when making meaningful plots of loss function geometry. We will explore the importance of proper scaling below as we explore the sharpness/flatness of different minimizers. In this context, we show that the sharpness of filter-normalized plots correlates with generalization error, while plots without filter normalization can be very misleading.

## 5    THE SHARP VS FLAT DILEMMA

Section 4 introduces the concept of filter normalization, and provides an intuitive justification for its use. In this section, we address the issue of whether sharp minimizers generalize better than flat minimizers. In doing so, we will see that the sharpness of minimizers correlates well with generalization error when filter normalization is used. This enables side-by-side comparisons between plots. In contrast, the sharpness of non-filter normalized plots may appear distorted and unpredictable.

It is widely thought that small-batch SGD produces "flat" minimizers that generalize better, while large batch sizes produce "sharp" minima with poor generalization (Chaudhari et al., 2017; Keskar et al., 2017; Hochreiter & Schmidhuber, 1997). This claim is disputed though, with Dinh et al. (2017); Kawaguchi et al. (2017) arguing that generalization is not directly related to the curvature of loss surfaces, and some authors proposing specialized training methods that achieve good performance with large batch sizes (Hoffer et al., 2017; Goyal et al., 2017; De et al., 2017).

Here, we explore the difference between sharp and flat minimizers. We begin by discussing difficulties that arise when performing such a visualization, and how proper normalization can prevent such plots from producing distorted results.

We train a CIFAR-10 classifier using a 9-layer VGG network (Simonyan & Zisserman, 2015) with Batch Normalization (Ioffe & Szegedy, 2015). We use two batch sizes: a large batch size of 8192 (16.4% of the training data of CIFAR-10), and a small batch size of 128. Let $\theta_s$ and $\theta_l$ indicate the solutions obtained by running SGD using small and large batch sizes, respectively[1]. Using the linear interpolation approach (Goodfellow et al., 2015), we plot the loss values on both training and testing data sets of CIFAR-10, along a direction containing the two solutions, i.e., $f(\alpha) = L(\theta_s + \alpha(\theta_l - \theta_s))$.

---

[1]In this section, we consider the "running mean" and "running variance" as trainable parameters and include them in $\theta$. Note that the original study by Goodfellow et al. (2015) does not consider batch normalization. These parameters are not included in $\theta$ in future sections, as they are only needed when interpolating between two minimizers.

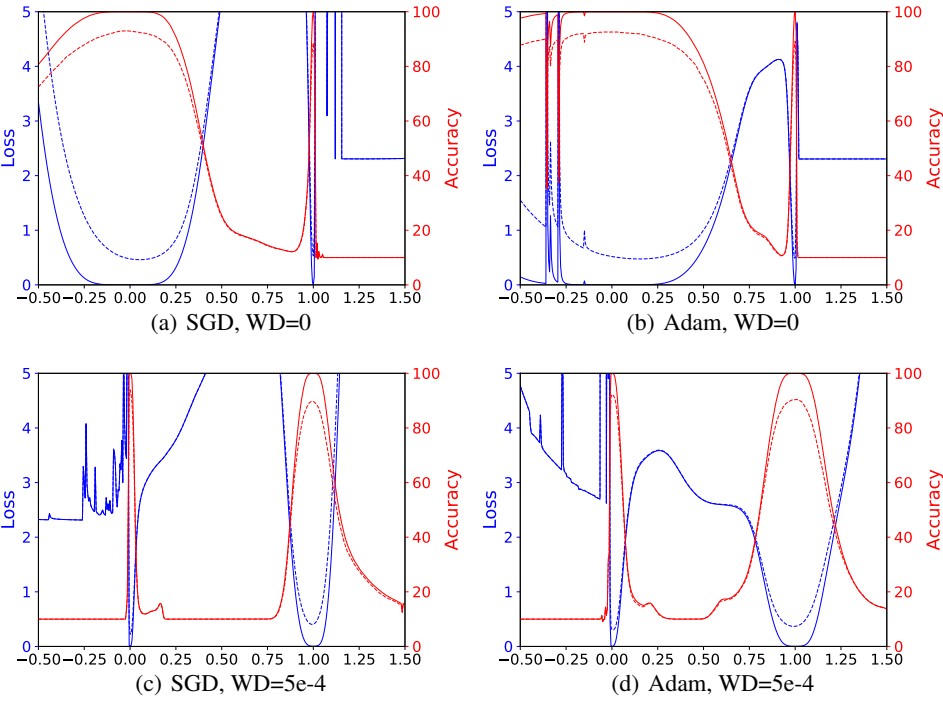

Figure 2: 1D linear interpolation of solutions obtained by small-batch and large-batch methods for VGG9. The blue lines are loss values and the red lines are accuracies. The solid lines are training curves and the dashed lines are for testing. Small batch is at abscissa 0, and large batch is at abscissa 1.

Table 1: Test errors of VGG-9 on CIFAR-10 with different optimization algorithms and hyper-parameters.

|  |  | SGD | | Adam | |
|---|---|---|---|---|---|
|  |  | bs=128 | bs=8192 | bs=128 | bs=8192 |
| VGG-9 | WD = 0 | 7.37 | 11.07 | **7.44** | 10.91 |
|  | WD = 5e-4 | **6.00** | **10.19** | 7.80 | **9.52** |

Similar to Keskar et al. (2017), we also superimpose the classification accuracy in red. This plot is shown in Figure 2.

Figures 2(a) and 2(b) show linear interpolation plots with $\theta_s$ at x-axis location 0, and $\theta_l$ at location $1^2$. As observed by Keskar et al. (2017), we can clearly see that the small-batch solution is quite wide, while the large-batch solution is sharp. However, this sharpness balance can be flipped simply by turning on weight decay (Krogh & Hertz, 1992). Figures 2(c) and 2(d) show results of the same experiment, except this time with a non-zero weight decay parameter. This time, the large batch minimizer is considerably flatter than the sharp small batch minimizer. However, we see from Table 1 that small batches generalize better in all 4 experiments; there is no apparent correlation between sharpness and generalization. We will see that these side-by-side sharpness comparisons are extremely misleading, and fail to capture the endogenous properties of the minima.

The apparent differences in sharpness in Figure 2 can be explained by examining the weights of each minimizer. Histograms of the networks weights are shown for each experiment in Figure 3. We see that, when a large batch is used with zero weight decay, the resulting weights tends to be smaller than in the small batch case. We reverse this effect by adding weight decay; in this case the large batch minimizer has much larger weights than the small batch minimizer. This difference in scale occurs for a simple reason: A smaller batch size results in more weight updates per epoch than a large batch size, and so the shrinking effect of weight decay (which imposes a penalty on the norm of the weights) is more pronounced.

Figure 2 in not visualizing the endogenous sharpness of minimizers, but rather just the (irrelevant) weight scaling. The scaling of weights in these networks is irrelevant because batch normalization re-scales the outputs to have unit variance. However, small weights still appear more sensitive to perturbations, and produce sharper looking minimizers.

**Filter normalized plots**   We repeat the experiment in Figure 2, but this time we plot the loss function near each minimizer separately using random filter-normalized directions. This removes the apparent differences in geometry caused by the scaling depicted in Figure 3. The results, presented in Figure 4, still show differences in sharpness between small batch and large batch minima, however these differences are much more subtle than it would appear in the un-normalized plots.

We also visualize these results using two random directions and contour plots. As shown in Figure 5, the weights obtained with small batch size and non-zero weight decay have wider contours than the sharper large batch minimizers. Similar for Resnet-56 appear in Figure 12 of the Appendix.

---

[2]The 1D interpolation method for plotting is described in detail in Section 3

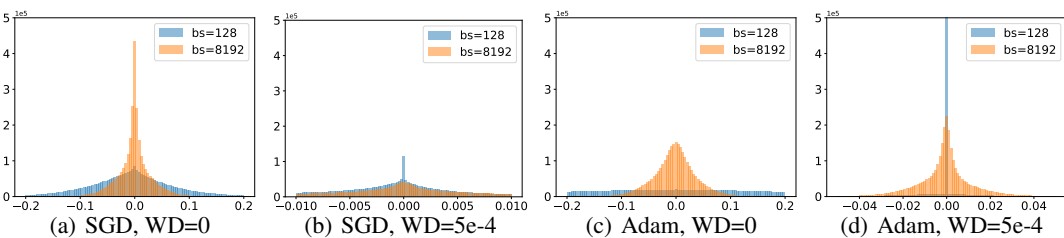

Figure 3: Histogram of weights. With zero weight decay, small-batch methods produce large weights. With non-zero weight decay, small-batch methods produce smaller weights.

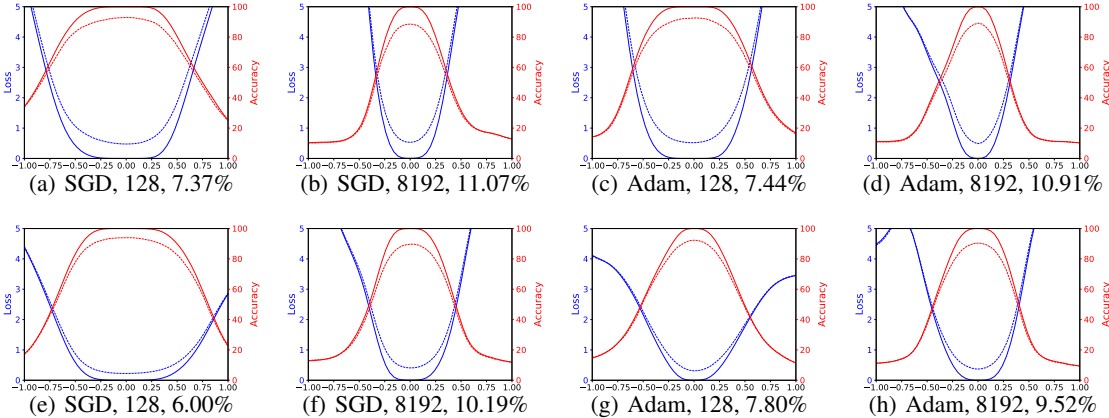

Figure 4: The shape of minima obtained using different optimization algorithms, with varying batch size and weight decay. The title of each subfigure contains the optimizer, batch size, and test error. The first row has no weight decay and the second row uses weight decay 5e-4.

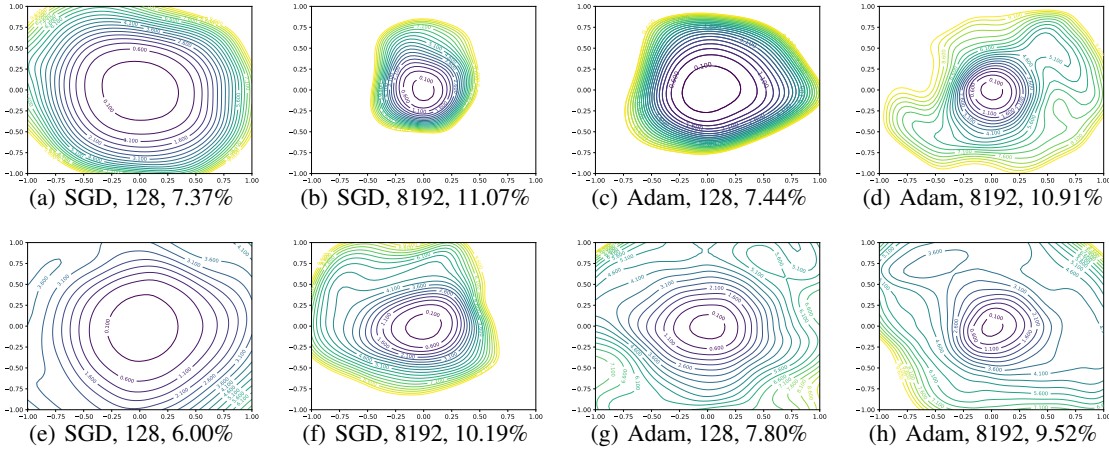

Figure 5: 2D visualization of solutions obtained by SGD with small-batch and large-batch. Similar to Figure 4, the first row uses zero weight decay and the second row sets weight decay to 5e-4.

**Generalization and Flatness**    Using the filter-normalized plots in Figures 4 and 5, we can make side-by-side comparisons between minimizers, and we see that now sharpness correlates well with generalization error. Large batches produced visually sharper minima (although not dramatically so) with higher test error. Interestingly, the Adam optimizer attained larger test error than SGD, and, as predicted, the corresponding minima are visually sharper. Results of a similar experiment using ResNet-56 are presented in the Appendix (Figure 12).

# 6    WHAT MAKES NEURAL NETWORKS TRAINABLE? INSIGHTS ON THE (NON) CONVEXITY STRUCTURE OF LOSS SURFACES

Our ability to find global minimizers to neural loss functions is not universal; it seems that some neural architectures are easier to minimize than others. For example, using skip connections, He et al. (2016) were able to train extremely deep architectures, while comparable architectures without skip connections are not trainable. Furthermore, our ability to train seems to depend strongly on the initial parameters from which training starts.

Using visualization methods, we do an empirical study of neural architectures to explore why the non-convexity of loss functions seems to be problematic in some situations, but not in others. We aim to provide insight into the following questions: Do loss functions have significant non-convexity at all? If prominent non-convexities exist, why are they not problematic in all situations? Why are some architectures easy to train, and why are results so sensitive to the initialization? We will see that different architectures have extreme differences in non-convexity structure that answer these questions, and that these differences correlate with generalization error.

## 6.1 EXPERIMENTAL SETUP

To understand the effects of network architecture on non-convexity, we trained a number of networks, and plotted the landscape around the obtained minimizers using the filter-normalized random direction method described in Section 4.

We consider three classes of neural networks:

- Residual networks that are optimized for performance on CIFAR (He et al., 2016). We consider ResNet-20, ResNet-56, and ResNet-110, where each name is labeled with the number of convolutional layers it has.

- "VGG-like" networks that do not contain shortcut/skip connections. We produced these networks simply by removing the skip connections from the CIFAR-optimized ResNets. We call these networks ResNet-20-noshort, ResNet-56-noshort, and ResNet-110-noshort. Note that these networks do not all perform well on the CIFAR-10 task. We use them purely for experimental purposes to explore the effect of shortcut connections.

- "Wide" ResNets that have been optimized for ImageNet rather than CIFAR. These networks have more filters per layer than the CIFAR optimized networks, and also have different numbers of layers. These models include ResNet-18, ResNet-34, and ResNet-50.

All models are trained on the CIFAR-10 dataset using SGD with Nesterov momentum, batch-size 128, and 0.0005 weight decay for 300 epochs. The learning rate was initialized at 0.1, and decreased by a factor of 10 at epochs 150, 225 and 275. Deeper experimental VGG-like networks (e.g., ResNet-56-noshort, as described below) required a smaller initial learning rate of 0.01.

High resolution 2D plots of the minimizers for different neural networks are shown in Figure 6. Results are shown as contour plots rather than surface plots because this makes it extremely easy to see non-convex structures and evaluate sharpness. For surface plots of ResNet-56, see Figure 1. Note that the center of each plot corresponds to the minimizer, and the two axes parameterize two random directions with filter-wise normalization as in (1). We make several observations below about how architecture effects the loss landscape. We also provide loss and error values for these networks in Table 2, and convergence curves in Figure 14 of the Appendix.

## 6.2 THE EFFECT OF NETWORK DEPTH

From Figure 6, we see that network depth has a dramatic effect on the loss surfaces of neural networks when skip connections are not used. The network ResNet-20-noshort has a fairly benign landscape dominated by a region with convex contours in the center, and no dramatic non-convexity. This isn't too surprising: the original VGG networks for ImageNet had 19 layers and could be trained effectively (Simonyan & Zisserman, 2015).

However, as network depth increases, the loss surface of the VGG-like nets spontaneously transitions from (nearly) convex to chaotic. ResNet-56-noshort has dramatic non-convexities and large regions where the gradient directions (which are normal to the contours depicted in the plots) do not point towards the minimizer at the center. Also, the loss function becomes extremely large as we move in some directions. ResNet-110-noshort displays even more dramatic non-convexities, and becomes extremely steep as we move in all directions shown in the plot. Furthermore, note that the minimizers at the center of the deep VGG-like nets seem to be fairly sharp. In the case of ResNet-56-noshort, the minimizer is also fairly ill-conditioned, as the contours near the minimizer have significant eccentricity.

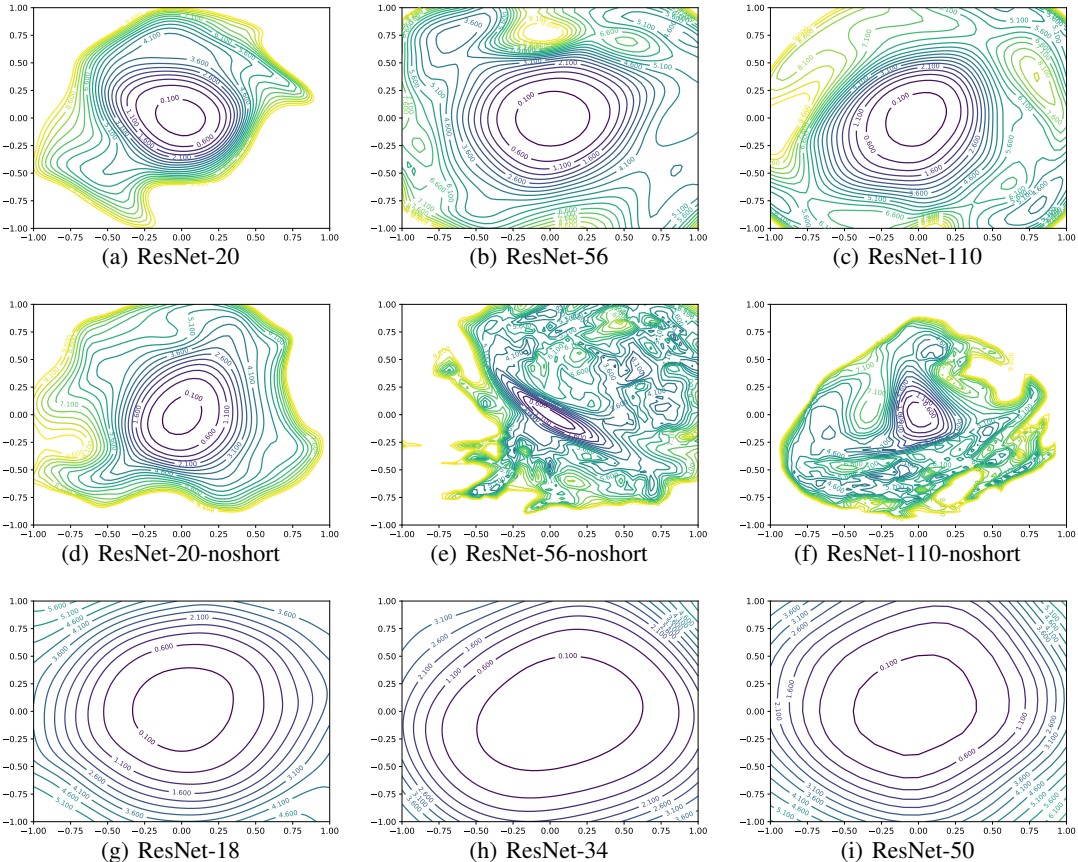

Figure 6: 2D visualization of the solutions of different networks.

Table 2: Loss values and errors for different architectures.

|  | Filters | Training Loss | Training Error | Test Error |
|---|---|---|---|---|
| ResNet-20 | 16 | 0.017 | 0.286 | 7.37 |
| ResNet-20-noshort |  | 0.025 | 0.560 | 8.18 |
| ResNet-56 | 16 | 0.004 | 0.052 | 5.89 |
| ResNet-56-noshort |  | 0.024 | 0.704 | 10.83 |
| ResNet-110 | 16 | 0.002 | 0.042 | 5.79 |
| ResNet-110-noshort |  | 0.258 | 8.732 | 16.44 |
| ResNet-18 | 64 | 0.002 | 0.026 | 5.42 |
| ResNet-34 | 64 | 0.001 | 0.014 | 4.73 |
| ResNet-50 | 64 | 0.001 | 0.006 | 4.55 |

## 6.3 SHORTCUT CONNECTIONS TO THE RESCUE

Shortcut connections have a dramatic effect of the geometry of the loss functions. In Figure 6, we see that residual connections prevent the transition to chaotic behavior as depth increases. In fact, the width and shape of the 0.1-level contour is almost identical for the 20- and 110-layer networks.

Interestingly, the effect of skip connections seems to be most important for deep networks. For the more shallow networks (ResNet-20 and ResNet-20-noshort), the effect of skip connections is fairly unnoticeable. However residual connections prevent the explosion of non-convexity that occurs when networks get deep. This effect seems to apply to other kinds of skip connections as well; Figure 13 of the Appendix shows the loss landscape of DenseNet (Huang et al., 2017), which shows no noticeable non-convexity.

## 6.4 WIDE MODELS VS THIN MODELS

To see the effect of the number of convolutional filters per layer, we compare the narrow CIFAR-optimized ResNets (ResNet-20/56/110) with wider ResNets (ResNet-18/34/50) that have more filters and were optimized for ImageNet. From Figure 6, we see that the wider models have loss landscapes with no noticeable chaotic behavior. Increased network width resulted in flat minima and wide regions of apparent convexity.

This effect is also validated by Figure 7, in which we plot the landscape of ResNet-56, but we multiple the number of filter per layer by $k = 2, 4$, and $8$. We see that increased width prevents chaotic behavior, and skip connections dramatically widen minimizers. Finally, note that sharpness correlates extremely well with test error.

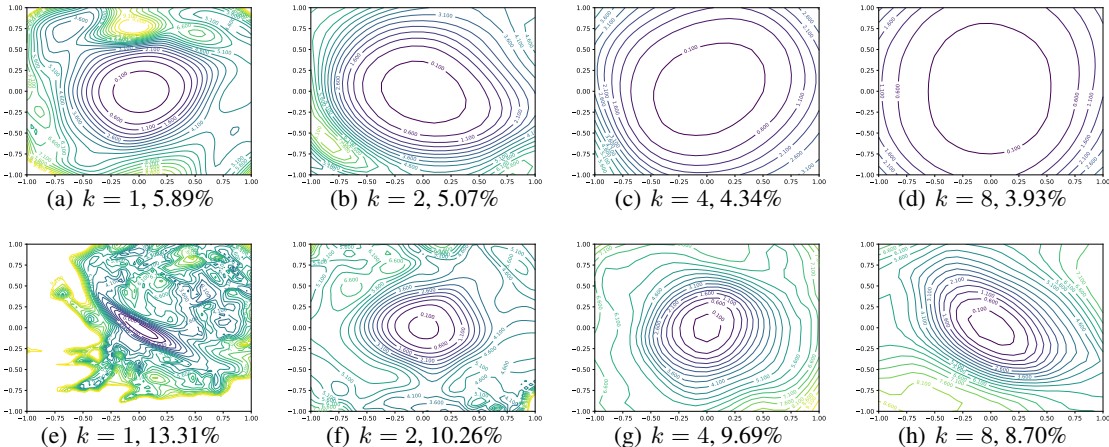

Figure 7: Wide-ResNet-56 (WRN-56) on CIFAR-10 both with shortcut connections (top) and without (bottom). The label $k = 2$ means twice as many filters per layer, $k = 4$ means 4 times, etc. Test error is reported below each figure.

## 6.5 IMPLICATIONS FOR NETWORK INITIALIZATION

One of the most interesting observations seen in Figure 6 is that loss landscapes for all the networks considered seem to be partitioned into a well-defined region of low loss value and convex contours, surrounded by a well-defined region of high loss value and non-convex contours.

This partitioning of chaotic and convex regions may explain the importance of good initialization strategies, and also the easy training behavior of "good" architectures. When using normalized random initialization strategies such as those proposed by Glorot & Bengio (2010), typical neural networks attain an initial loss value less than 2.5. The well behaved loss landscapes in Figure 6 (ResNets, and shallow VGG-like nets) are dominated by large, flat, nearly convex attractors that rise to a loss value of 4 or greater. For such landscapes, a random initialization will likely lie in the "well- behaved" loss region, and the optimization algorithm might never "see" the pathological non-convexities that occur on the high loss chaotic plateaus.

Chaotic loss landscapes (ResNet-56-noshort and ResNet-110-noshort) have shallower regions of convexity that rise to lower loss values. For sufficiently deep networks with shallow enough attractors, the initial iterate will likely lie in the chaotic region where the gradients are uninformative. In our experiments, SGD was unable to train a 156 layer network without skip connections (even with very low learning rates), which adds weight to this hypothesis.

## 6.6 LANDSCAPE GEOMETRY AFFECTS GENERALIZATION

Table 2 displays the training and test error for the networks depicted in Figure 6. Both Figures 6 and 7 show that landscape geometry has a dramatic effect on generalization. First, note that visually

flatter minimizers consistently correspond to lower test error, which further strengthens our assertion that filter normalization is a natural way to visualize loss function geometry.

Second, we notice that chaotic landscapes (deep networks without skip connections) result in worse training and test error, while more convex landscapes have lower error values. In fact, the most convex landscapes (the wide ResNets in the bottom row of Figure 6) generalize the best of all the networks. This latter class of networks show no noticeable chaotic behavior at all.

# 7 VISUALIZING OPTIMIZATION PATHS

Finally, we explore methods for visualizing the trajectories of different optimizers. For this application, random directions are ineffective. We will provide a theoretical explanation for why random directions fail, and explore methods for effectively plotting trajectories on top of loss function contours.

Several authors have observed that random direction fail to capture the variation in optimization trajectories, including Gallagher & Downs (2003); Lorch (2016); Lipton (2016); Liao & Poggio (2017). Several failed visualizations are depicted in Figure 8. In Figure 8(a), we see the iterates of SGD projected onto the plane defined by two random directions. Almost none of the motion is captured (notice the super-zoomed-in axes and the seemingly random walk). This problem was noticed by Goodfellow et al. (2015), who then visualized trajectories using one direction that points from initialization to solution, and one random direction. This approach is shown in Figure 8(b). As seen in Figure 8(c), the random axis captures almost no variation, leading to the (misleading) appearance of a straight line path.

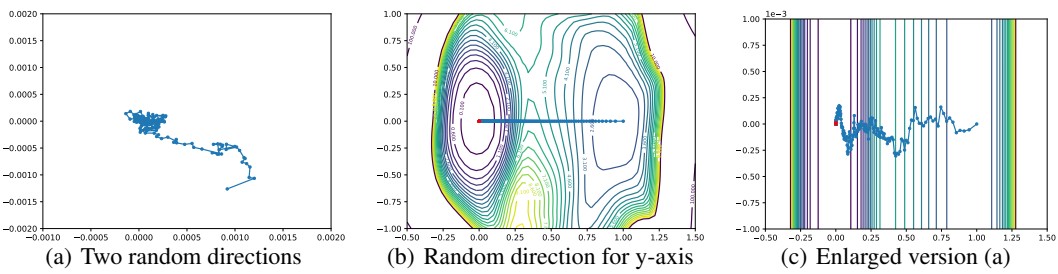

(a) Two random directions     (b) Random direction for y-axis     (c) Enlarged version (a)

Figure 8: Ineffective visualizations of optimizer trajectories. These visualizations suffer from the orthogonality of random directions in high dimensions.

## 7.1 WHY RANDOM DIRECTIONS FAIL: LOW DIMENSIONAL OPTIMIZATION TRAJECTORIES

It is well known that two random vectors in a high dimensional space will be nearly orthogonal with high probability. In fact, the expected cosine similarity between Gaussian random vectors in $n$ dimensions is roughly $\sqrt{2/(\pi n)}$ (Goldstein & Studer (2016), Lemma 5).

This is problematic when optimization trajectories lie in extremely low dimensional spaces. In this case, a randomly chosen vector will lie orthogonal to the low-rank space containing the optimization path, and a projection onto a random direction will capture almost no variation. Figure 8(b) suggests that optimization trajectories are low dimensional because the random direction captures orders of magnitude less variation than the vector that points along the optimization path. Below, we use PCA directions to directly validate this low dimensionality, and also to produce effective visualizations.

## 7.2 EFFECTIVE TRAJECTORY PLOTTING USING PCA DIRECTIONS

To capture variation in trajectories, we need to use non-random (and carefully chosen) directions. Here, we suggest an approach based on PCA that allows us to measure how much variation we've captured; we also provide plots of these trajectories along the contours of the loss surface.

Let $\theta_i$ denote model parameters at epoch $i$ and the final solution as $\theta_n$. Given $m$ training epochs, we can apply PCA to the matrix $M = [\theta_0 - \theta_n; \cdots ; \theta_{n-1} - \theta_n]$, and then select the two most

explanatory directions. Optimizer trajectories (blue dots) and loss surfaces along PCA directions are shown in Figure 9. Epochs where the learning rate was decreased are shown as red dots. On each axis, we measure the amount of variation in the descent path captured by that PCA direction.

We see some interesting behavior in these plots. At early stages of training, the paths tend to move perpendicular to the contours of the loss surface, i.e., along the gradient directions as one would expect from non-stochastic gradient descent. The stochasticity becomes fairly pronounced in several plots during the later stages of training. This is particularly true of the plots that use weight decay and small batches (which leads to more gradient noise, and a more radical departure from deterministic gradient directions). When weight decay and small batches are used, we see the path turn nearly parallel to the contours and "orbit" the solution when the stepsize is large. When the stepsize is dropped (at the red dot), the effective noise in the system decreases, and we see a kink in the path as the trajectory falls into the nearest local minimizer.

Finally, we can directly observe that the descent path is very low dimensional: between 40% and 90% of the variation in the descent paths lies in a space of only 2 dimensions. The optimization trajectories in Figure 9 appear to be dominated by movement in the direction of a nearby attractor. This low dimensionality is compatible with the observations in Section 6.5, where we observed that non-chaotic landscapes are dominated by wide, flat minimizers.

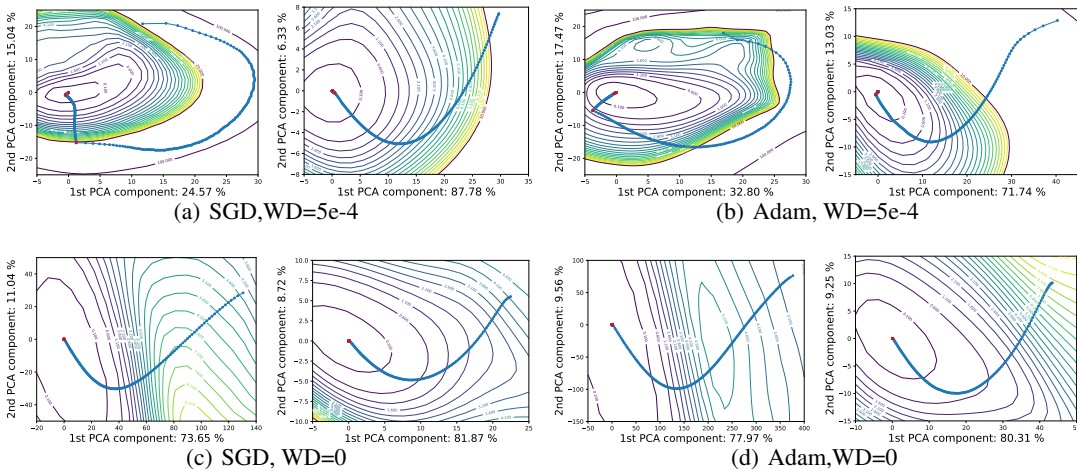

Figure 9: Projected learning trajectories use normalized PCA directions for VGG-9. The left plot in each subfigure uses batch size 128, and the right one uses batch size 8192.

## 8 CONCLUSION

In this paper, we presented a new, more accurate visualization technique that provided insights into the consequences of a variety of choices facing the neural network practitioner, including network architecture, optimizer selection, and batch size.

Neural networks have advanced dramatically in recent years, largely on the back of anecdotal knowledge and theoretical results with complex assumptions. For progress to continue to be made, a more general understanding of the structure of neural networks is needed. Our hope is that effective visualization, when coupled with continued advances in theory, can result in faster training, simpler models, and better generalization.

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

# 9 APPENDIX

## 9.1 LARGE-BATCH AND SMALL-BATCH RESULTS FOR RESNET-56

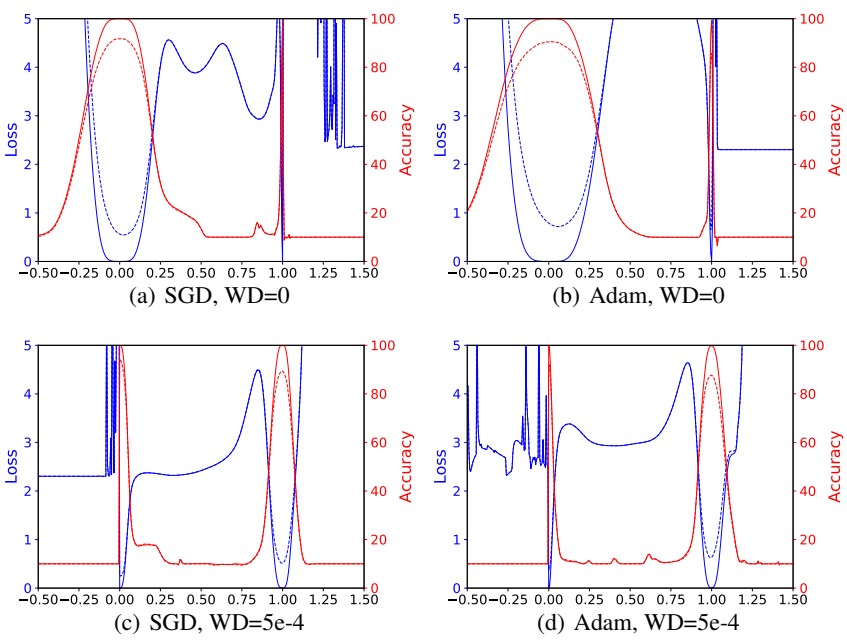

Figure 10: 1D linear interpolation of solutions obtained by small-batch and large-batch methods for ResNet56. The blue lines are loss values and the red lines are error. The solid lines are training curves and the dashed lines are for testing.

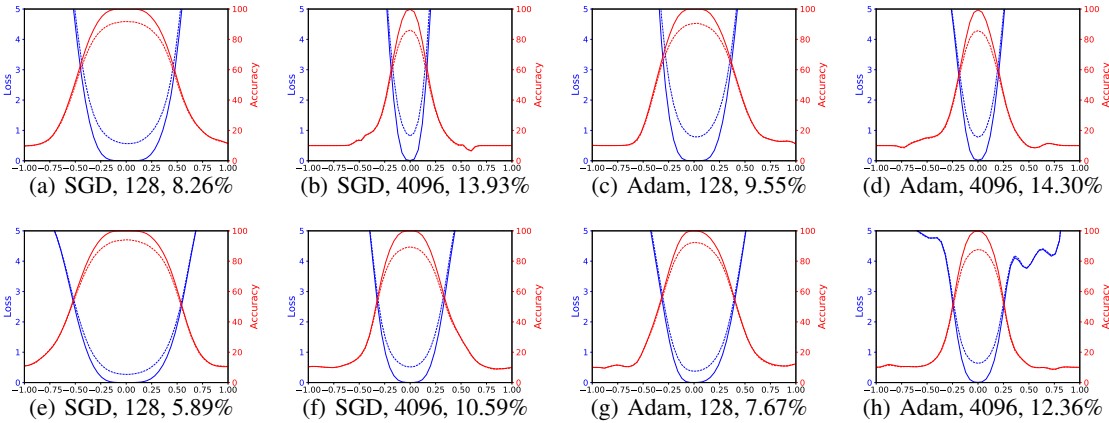

Figure 11: The shape of minima obtained via different optimization algorithms for ResNet-56, with varying batch size and weight decay. Similar to Figure 4, the first row uses zero weight decay and the second row uses 5e-4 weight decay.

Generalization error for each plot is shown in Table 3.

# 10 TEST AND TRAINING DATA FOR VARIOUS NETWORKS

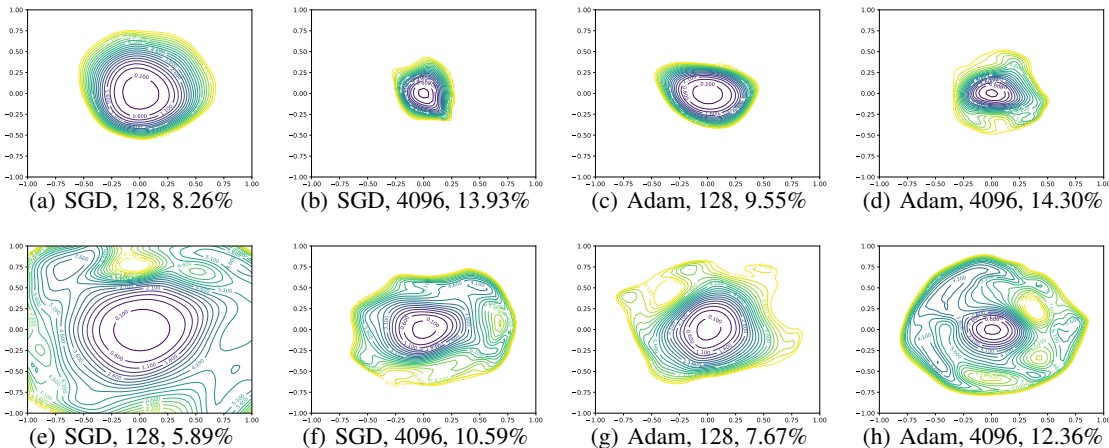

(a) SGD, 128, 8.26%     (b) SGD, 4096, 13.93%     (c) Adam, 128, 9.55%     (d) Adam, 4096, 14.30%

(e) SGD, 128, 5.89%     (f) SGD, 4096, 10.59%     (g) Adam, 128, 7.67%     (h) Adam, 4096, 12.36%

Figure 12: 2D visualization of solutions of ResNet-56 obtained by SGD/Adam with small-batch and large-batch. Similar to Figure 11, the first row uses zero weight decay and the second row sets weight decay to 5e-4.

Table 3: Test error for ResNet-56 with different optimization algorithms and batch-size/weight-decay parameters.

|            | SGD | | Adam | |
|------------|--------|---------|--------|---------|
|            | bs=128 | bs=4096 | bs=128 | bs=4096 |
| WD = 0     | 8.26   | 13.93   | 9.55   | 14.30   |
| WD = 5e-4  | **5.89** | **10.59** | **7.67** | **12.36** |

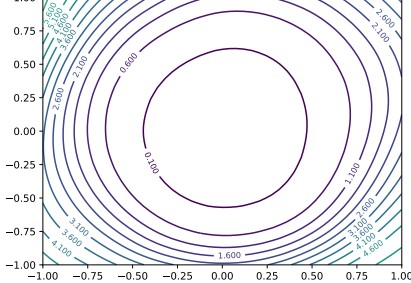

Figure 13: The loss landscape for DenseNet-121 trained on CIFAR-10. The final training error is 0.002 and the testing error is 4.37

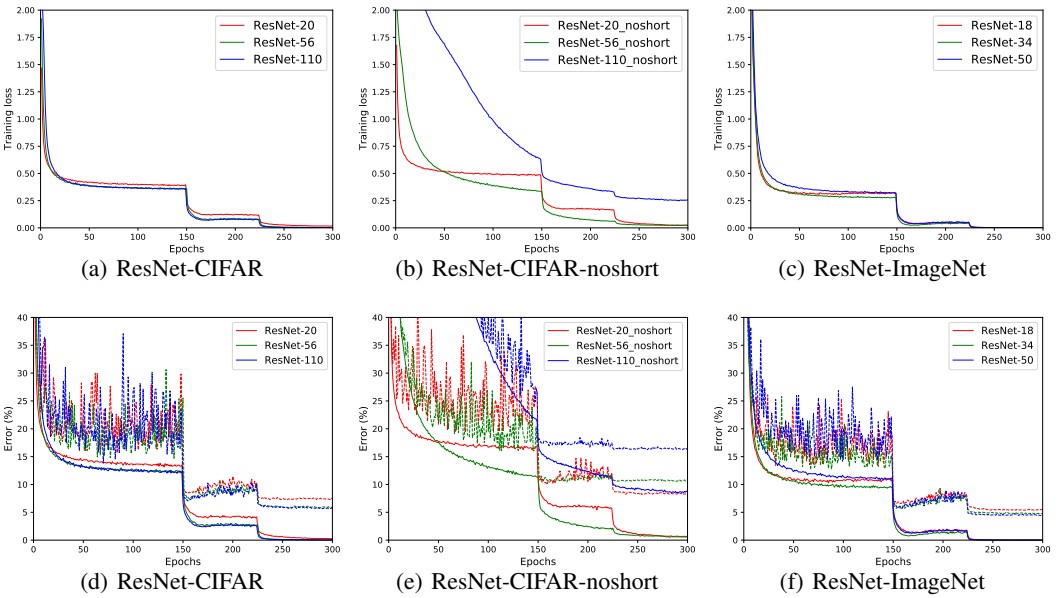

Figure 14: Convergence curves for different architectures. The first row is for training loss and the second row are training and testing error curves.

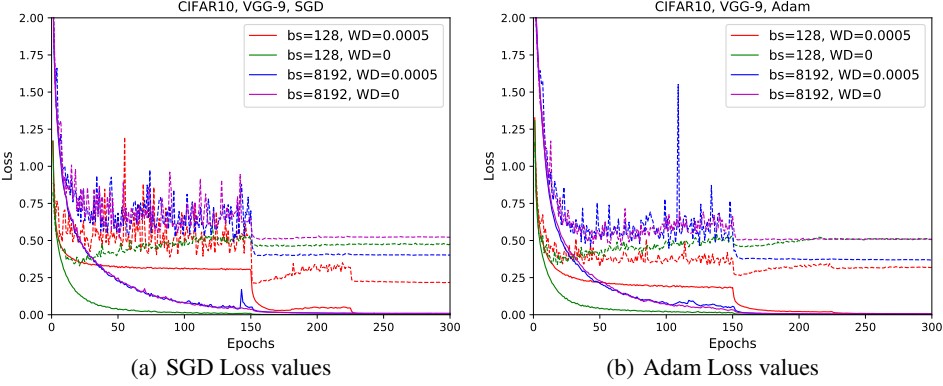

Figure 15: Training and testing loss curves for VGG-9. Dashed lines are for testing, solid for training.

