# OpenReview forum: "Visualizing the Loss Landscape of Neural Nets"
_ICLR.cc/2018/Conference — Invite to Workshop Track_

### Official Review · AnonReviewer1 · 2017-11-27
**Nice visualizations but conclusions are unclear**

**Rating:** 5
**Confidence:** 4

**Review:**

This paper provides visualizations of different deep network loss surfaces using 2D contour plots, both at minima and along optimization trajectories. They mention some subtle details that must be taken into account, such as scaling the plot axes by the filter magnitudes, in order to obtain correctly scaled plots.

Overall, I think there is potential with this work but it feels preliminary. The visualizations are interesting and provide some general intuition, but they don't yield any clear novel insights that could be used in practice. Also, several parts of the paper spend too much time on describing other work or on implementation details which could be moved to the appendix.

General Comments:
- I think Sections 2, 3, 4 are too long, we only start getting to the results section at the end of page 4. I suggest shortening Section 2, and it should be possible to combine Sections 3 and 4 into a page at most. 1D interpolations and 2D contour plots can be described in a few sentences each.
- I think Section 5 can be put in the Appendix - it's essentially an illustration of why the weight scaling is important. Once these details are done correctly, the experiments support the relatively well-accepted hypothesis that flat minima generalize better.
- The plots in Section 6 are interesting, it would be nice if the authors had an explanation of why the loss surface changes the way it does when skip connections are added.
- In Section 7, it's less useful to spend time describing what happens when the visualization is done wrong (i.e. projecting along random directions rather than PCA vectors) -  this can be put in the Appendix. I would suggest just including the visualizations of the optimization trajectories which are done correctly and focus on deriving interesting/useful conclusions from them.

---

> ### Author Response · Authors · 2017-12-24
> **Response to comments**
>
> We thank the reviewer for the kind feedback and constructive suggestions.  We agree with the reviewer that some sections could be shortened or moved to appendix and more efforts should be focused on the interpretation of results.  We have made major revisions to the paper to address these issues.  In particular, we shortened the first 3 sections of the paper, and we added several discussions into Section 6 that specifically address ramifications of our findings, and how loss landscape geometry effects trainability and generalization error.
>
> Finally, please see Section 1.1 of the new draft, which lists our contributions.  We think there are a number of new discoveries in this paper (in particular our realizations about the transition between convex and chaotic landscapes) that the reviewer may have overlooked.  We have done a lot of writing to change the focus of our paper to analyze in detail the observations we make about our visualizations.
>
> The reviewer also seems concerned that this paper is overly long. This is largely due to the number of figures. In fact, we have nearly 4 pages of figures (and about 8 pages of text, which is on par with the suggested length).  This is a paper on visualization methods, and as a result it’s hard to chop down on these space consuming figures without losing important content.
>
> We answer a few of the reviewer’s direct questions below.
>
> Q1: I think Sections 2, 3, 4 are too long, we only start getting to the results section at the end of page 4. I suggest shortening Section 2, and it should be possible to combine Sections 3 and 4 into a page at most. 1D interpolations and 2D contour plots can be described in a few sentences each.
> A:  We have shortened Sections 3 and 4 to 1.5 pages total.  We think there is some important discussion to be had here, in particular to justify the reasoning for the filter normalization.  Unfortunately, not all readers will be familiar with issues like scale invariance, batch normalization, and various plotting methods. These methods form the foundation for the paper, so we don’t want to gloss over these too lightly.
>
> Q2: I think Section 5 can be put in the Appendix - it's essentially an illustration of why the weight scaling is important. Once these details are done correctly, the experiments support the relatively well-accepted hypothesis that flat minima generalize better.
> A:  There are two reasons for including Section 5:  First, we reveal that much of the work documenting that flat minimizers are better is actually false of misleading.  Several other authors have noted this, and some even claim to have refuted the sharp vs flat hypothesis (see Dinh, 2017, “Sharp Minima Can Generalize for Deep Nets”).
>
> Second, it is important to validate that filter normalization produces plots with sharpness that actually correlates with generalization error.  Without Section 5, there would be no validation of the accuracy of our method for comparing loss functions.

---

### Official Review · AnonReviewer3 · 2017-11-28
**The submission considers the problem of visualizing loss functions of NNs and provides some interesting insights on the trainability and the generalization of NNs. However, it seems its novelty is quite limited.**

**Rating:** 4
**Confidence:** 3

**Review:**

The main concern of this submission is the novelty. Proposed method to visualize the loss function sounds too incremental from existing works. One of the main distinctions is using filter-wise normalization, but it is somehow trivial. In experiments, no comparisons against existing works is performed (at least on toy/controlled environments). Some findings in this submission indeed look interesting, but it is not clear if those results are something difficult to find with other existing standard ways, or even how reliable they are since the effectiveness has not been evaluated.

Minor comments:
In introduction, parameter with zero training error doesn't mean it's a global minimizer
In section 2, it is not clear that visualizing loss function is helpful in see the reasons of generalization given minima.
In figure 2, why do we have solutions at 0 for small batch size and 1 for large batch size case? (why should they be different?)

---

> ### Author Response · Authors · 2017-12-24
> **Defending the contributions and response to questions**
>
> We thank the reviewer for the valuable feedback. The main contribution of this paper is not the filter normalization scheme itself, but rather the first thorough empirical investigation of neural loss functions. While the filter normalization scheme is also a contribution, it is merely a means to an end; it enables us to plot different loss functions and minimizers on a normalized scale so they can be compared side-by-side. Loss function visualizations reveal a number of important things that have not been observed in the literature. This includes the transition between smooth and chaotic loss landscapes with increased network depth, the important role that these qualitative differences play in generalization error, and the dramatic effect of skip connections of loss function structure. We have added a section on our contributions (Section 1.1) to help the reader navigate this paper.
>
> Q1: “One of the main distinctions is using filter-wise normalization, but it is somehow trivial. In experiments, no comparisons against existing works is performed..."
> A:  We make fairly extensive comparisons against an existing and commonly used method (linear interpolation).  We feel that Section 5 makes a convincing argument for why filter normalization advances the state of the art; without it, one cannot make meaningful sharpness vs flatness comparisons between different minima. This is demonstrated in Fig 2 and Table 1, which show that side-by-side comparisons of minima are not meaningful when linear interpolation (the current STOA) is used, and Fig. 4 which shows that filter normalization makes sharpness correlate with generalization error.  We validate this using network architectures, different optimizers, and different optimization parameters (batch size and weight decay).
>
> While the filter normalization scheme is indeed quite simple (which we view as a merit), it yields a nontrivial improvement over existing methods. We think this observation is significant because of the pervasive use of linear interpolation methods to visualize sharpness, which we show to produce misleading results due to the scaling effect.
>
> Finally, we note that filter normalization is only one of many contributions of this paper .  Please see Section 1.1 in the new draft, which lists our contributions.
>
> Q2: “it is not clear if those results are something difficult to find with other existing standard ways, or even how reliable they are since the effectiveness has not been evaluated.”
> A: Section 5 shows that loss surfaces cannot be compared meaningfully without filter normalization, and that loss surface sharpness with filter normalization correlates with generalization error for a range of different architectures and training methods. Also, this is the first article to present high resolution visualizations of loss functions that reveal the dramatic qualitative differences between network architectures. We think this is a major contribution of the paper, and the significance of this result does not depend on the novelty of filter normalization (which is merely a tool for making side-by-side comparisons of sharpness between different plots).
>
> Q3: “In introduction, parameter with zero training error doesn't mean it's a global minimizer”
> A:  Thanks for pointing out the typo. We mean zero training “loss” not “error’’.  Since cross-entropy loss is non-negative, any zero loss minimizer is a global minimizer.
>
> Q4: “In section 2, it is not clear that visualizing loss function is helpful in see the reasons of generalization given minima.“
> A:   Section 2 is meant to review theoretical results on the structure of loss functions.  Later in the paper, we investigate two ways in which loss characteristics affect generalization, and both of these characteristics are easily explored via visualization. In Section 5, we show that the sharpness of filter-normalized plots correlates with generalization error. In Section 6, we also show that chaotic loss landscape geometry also results in poor generalization. We add Section 6.5 which discusses how loss function geometry effects initialization, and reasons why it is not possible to train neural networks effectively once loss landscapes get sufficiently chaotic.
>
> Q5: “In figure 2, why do we have solutions at 0 for small batch size and 1 for large batch size case? (why should they be different?)”
> A:  We use the same setting as Keskar et. al, 2017, which compare the small/large-batch solutions using the linear interpolation method. Given two solutions trained with different batch size, \theta_s and \theta_l, we can linearly interpolate them using the formula (1-\alpha)*\theta_s + \alpha*\theta_l. For each value of \alpha, we compute the loss function for the corresponding interpolated parameters. The plots in Figure 2 have \alpha on the x-axis . When \alpha=0, this is the loss of \theta_s, and when \alpha=1, this is the loss of \theta_l.

---

### Official Review · AnonReviewer2 · 2017-11-29
**Throughout visualisation, this paper investigates the "flat vs sharp dilemma", the non convexity of the loss surface and the so-called optimisation paths. Nice plots but I would have appreciated a deeper treatment of observations.**

**Rating:** 5
**Confidence:** 3

**Review:**


* In the "flat vs sharp" dilemma, the experiments display that the dilemma, if any, is subtle. Table 1 does not necessarily contradict this view. It would be a good idea to put the test results directly on Fig. 4 as it does not ease reading currently (and postpone ResNet-56 in the appendix).

How was Figure 5 computed ? It is said that *a* random direction was used from each minimiser to plot the loss, so how the 2D directions obtained ?

* On the convexity vs non-convexity (Sec. 6), it is interesting to see how pushing the Id through the net changes the look of the loss for deep nets. The difference VGG - ResNets is also interesting, but it would have been interesting to see how this affects the current state of the art in understanding deep learning, something that was done for the "flat vs sharp" dilemma, but is lacking here. For example, does this observation that the local curvature of the loss around minima is different for ResNets and VGG allows to interpret the difference in their performances ?

* On optimisation paths, the choice of PCA directions is wise compared to random projections, and results are nice as plotted. There is however a phenomenon I would have liked to be discussed, the fact that the leading eigenvector captures so much variability, which perhaps signals that optimisation happens in a very low dimensional subspace for the experiments carried, and could be useful for optimisation algorithms (you trade dimension d for a much smaller "effective" d', you only have to figure out a generating system for this subspace and carry out optimisation inside). Can this be related to the "flat vs sharp" dilemma ? I would suppose that flatness tends to increase the variability captured by leading eigenvectors ?


Typoes:

Legend of Figure 2: red lines are error -> red lines are accuracy
Table 1: test accuracy -> test error
Before 6.2: architecture effects -> architecture affects

---

> ### Author Response · Authors · 2017-12-24
> **Response to comments and questions**
>
> We thank the reviewer for the valuable feedback and constructive suggestions. Here are our thoughts on the comments:
>
> Q1: “In the "flat vs sharp" dilemma, the experiments display that the dilemma, if any, is subtle. Table 1 does not necessarily contradict this view”.
> A:    The purpose of Table 1 is to contradict the notion that 1D linear interpolation is a meaningful view of sharpness/flatness.  Please examine Fig. 2 in our paper. The top two figures show small batches producing flatter minimizers and Table 1 shows that flat minimizers produce good generalization. However, the bottom 2 figures (with weight decay) reverse this result, in which the large batch solutions produce “flatter minimizers,” even though these minimizers have worse generalization error than the “sharp” looking small-batch minimizers. In other words, the apparent sharpness/flatness of 1D interpolation is easily manipulated, and does not correspond to generalization.
>
> The problem we have revealed is that 1D linear interpolation is predominantly visualizing the scale of the weights rather than the endogenous sharpness/flatness. We show that the filter-normalized view is a more reliable way to make visual comparisons of the sharpness among minima. With filter normalization, flatness of the resulting visualizations corresponds to increased generalization ability.
>
> Finally, we note that the differences in sharpness/flatness in Figure 4 are indeed subtle.  We view this as one of our contributions:  previous work using 1D interpolation has depicted these differences as being extremely dramatic, but we show that these dramatic differences are largely a distortion caused by differences in weight scaling.
>
> We have revised section 5 to make our contributions, and the purpose of the figures, more clear.
>
> Q2: “It would be a good idea to put the test results directly on Fig. 4 as it does not ease reading currently”
> A: Great idea - we agree and we have added the test errors under each subfigure for easier comparison.
>
> Q3: “How was Figure 5 computed ? It is said that *a* random direction was used from each minimiser to plot the loss, so how the 2D directions obtained ?”
> A: To plot the 2D contours, we choose two random directions (say, a and b) and normalize them at the filter level. This means that, for each convolutional filter in the network, the corresponding entries in “a” contain a random (Gaussian) vector with the same dimensions and the same norm as that filter.   For each point (\alpha, \beta) in the figure, we calculate the loss value L(\theta + \alpha * a + \beta * b). We described the method of plotting 2D contours in section 3, and we will clarify it in section 5.  We have also added equation (1) in the new draft, which clarifies how these plots are made.
>
> Q4: “it would have been interesting to see how this affects the current state of the art in understanding deep learning, something that was done for the "flat vs sharp" dilemma...”
> A:  We think the observations in Section 6 say a lot about why certain networks perform better than others.  The local curvature around minima is very helpful in interpreting/explaining the performance difference between ResNets and VGG-like networks.
>    The 2D plots in Section 6 go beyond sharp vs flat, and reveal another important phenomenon that seems to have gone unnoticed in the literature; as network depth increases, loss landscapes suddenly transition from being smooth and dominated by nearly-convex regions, to being chaotic and highly non-convex.  Interestingly, the neural nets with smooth convex-like landscapes have low generalization error, whereas the chaotic landscapes yield high error.  We can improve the generalization of deep nets by taking measures to convexify loss landscape.  Skip connections preserve smoothness for deeper networks, and we see that these convex-like landscapes produce low error (Table 2).  Another approach is to widen the network, which also preserves smoothness for deeper networks.
>
> To address the issue raised by the reviewer, we have re-written Sections 6.2-6.4, and added Sections 6.5 and 6.6, which discuss the issues of generalization error and trainability.  We will also label the plots in Figure 6 to show the generalization error.
>
> Q5: “Can the fact that the leading eigenvector captures so much variability be related to the "flat vs sharp" dilemma ?”
> A: Good question. One striking thing that we can observe with confidence is that the high amount of variability captured using only 2 dimensions (sometimes as high a 90% for both dimensions combines) indicates that optimization trajectories lie in a very low dimensional space. This could be because well-behaved loss landscapes have large, flat, nearly-convex structures, and iterates move predominantly in the direction towards the nearest minimizer. To address the reviewers question, we have added a discussion of this at the end of Section 7.2.

---

### Public Comment · (anonymous) · 2017-12-15
**Reproduction Attempt**

We attempted to re-implement a subset of the results of this paper as part of the ICLR 2018 Reproducibility Challenge (http://www.cs.mcgill.ca/~jpineau/ICLR2018-ReproducibilityChallenge.html) in order to report on the feasibility of reproducing the authors’ findings. The authors described three different sets of experiments (small vs. large batch minimizer sharpness, convexity structure of loss surfaces, and optimizer trajectory visualization ).
The following is a breakdown of our approach and some comments regarding each of the sections we attempted to re-implement. Our approach was mainly limited by external factors such as time constraints and memory/machine requirements. The reproduction of this paper would have been more accurate and less time-consuming had we been provided the code used to generate the various figures and underlying network architectures. Moreover, we believe it would have been useful to have known the computing resources used to train the nets and generate the figures as well as an estimate of the total time to do so.

SECTION 5: We trained a 9-layer VGG based off of the previous work referenced in the paper. We were unable to use a large batch-size of 8192 and had to reduce to a  batch-size 512, which most likely impacted our results. We used any other hyperparameters specified by the authors. We did not note a particularly large difference between the sharpness/flatness of minima in the regular linear-interpolation vs. filter-normalized graphs, but this likely had to do with the significantly smaller large-batch size we ended up using. Moreover, although stating that the VGG9 network used to pilot the experiments had batch regularization, it is not clear where in the network these layers should be used (i.e after every convolution layer? Every layer?) and whether or not the weights stored in the batch normalization layer should be taken into account when generating the high dimensional gaussian vectors and if they should be used in the proposed filter-wise normalisation technique. Finally, the architecture of the VGG9 network doesn’t seem to be part of one of predefined reference models used by the authors who create them. As such, we had to make guesses as to how many convolution layers should be used before max pooling the their outputs. We also assumed the proposed default settings for the Adam optimizer for beta  1 and 2 as well as for epsilon from the original paper.
We noted a few minor typos, omissions, or otherwise unclear aspects that somewhat hindered our attempt.
-        In the description of the filter-wise normalization formula of section 4, “d_{f}” represents the i_th filter of d—shouldn’t it be “d_{i}”? Additionally, is ||d_{i}|| to be interpreted as the Frobenius norm as well?
-        There are some ambiguities regarding various figure captions and axis labels. The caption of figure 2 states the red lines represent error rather than accuracy. Similarly, the caption of table 1 describes its contents as accuracy rather than error.
- 	In Figure 3, the number of bins used to generate the histogram was not specified as well as whether or not we should count the weights from the batch normalization layer(s).  If given the total number of weights (i.e |𝜭|), this could have given us an insight on how to better generate / replicate the graphs.
-        We were unsure of what the x-axis is representing in figure 4, as it isn’t labelled. We assumed it represented alpha, the parameter used to scale the random high-dimensional gaussian vector , and a vector approach to generate other graphs with which we evaluate the accuracy. Similarly, we assumed the same for Figure 5 on both the x and y axes.

SECTION 6: We only re-implemented a portion of the experiments in this section, since we were limited in time and computing power. We didn’t attempt to re-implement “Wide” ResNets as they require fine tuning on the ImageNet dataset which we were unable to train on due to space and computing constraints. We attempted to use the original caffe model for normal CIFAR-10 ResNet but it proved incompatible with our hardware and thus resorted to the official Tensorflow implementation which we were able to train for sizes 20, 56, 110. Unfortunately extracting loss contour plots out of these off-the-shelf models proved to be difficult and required lengthy computations so we chose to prioritize the VGG visualizations.
Some general observations:
-        It is not specified whether training or testing losses were used to generate the graphs in figure 5.
-	For Figures  2,3,4,5,6,7,8, the step-sizes for the alpha and beta values of the different gaussian vectors to generate the mesh grid is not specified. This could, in turn, affect the resolution, or granularity, of our generated figures.

---

> ### Author Response · Authors · 2017-12-21
> **Thanks for the reproducing efforts and detailed comments**
>
> Thanks for your interests in our work and the efforts of reproducing the results! We really appreciate the detailed comments and suggestions. We will add more details in our next version soon and here we would like to clarify some missing details:
>
> Q1: “we believe it would have been useful to have known the computing resources used to train the nets and generate the figures as well as an estimate of the total time to do so.”
> A: We will add descriptions about the computing resources and the estimated time. Our PyTorch code can be executed in a multiple GPU workstation as well as a HPC with hundreds of GPUs using mpi4py. The computation time depends on the model’s inference speed on the training set, the resolution of the plots and the number of GPUs. For example, a 2D contour of ResNet-56 model with a (relatively low) resolution of 51x51 will take about 1 hour on a workstation with 4 GPUs (Titan X Pascal or 1080 Ti).  A much higher resolution version could take 64 GPUs 4 days.
>
> Q2: “whether or not the weights stored in the batch normalization layer should be taken into account when generating the high dimensional gaussian vectors and if they should be used in the proposed filter-wise normalisation technique”
> A: In the 1D linear interpolation methods, the BN parameters including the “running mean” and “running variance” need to be considered as part of \theta. If these parameters are not considered, then it is not possible to reproduce the loss accurately for both minimizers.  In the filter-normalized visualization, the random direction applies to all weights but not the weights in BN. Note that the filter normalization process removes the effect of weight scaling, and so the batch normalization can be ignored.
>
> Q3: The VGG-9 architecture details and parameters for Adam
> A: VGG-9 is a cropped version of VGG-16, which keeps the first 7 Conv layers in VGG-16 with 2 FC layers. A BN layer is added after each conv layer and the first FC layer. A detailed description of VGG-9 architecture can be also found in https://arxiv.org/pdf/1706.02379.pdf. We find VGG-9 is an efficient network with better performance comparing to VGG-16 on CIFAR-10. We use the default values for betas and epsilon in Adam (http://pytorch.org/docs/master/optim.html#torch.optim.Adam) with the same learning rate schedule as used in SGD.
>
> Q4: “In the description of the filter-wise normalization formula of section 4, “d_{f}” represents the i_th filter of d—shouldn’t it be “d_{i}”? Additionally, is ||d_{i}|| to be interpreted as the Frobenius norm as well?”
> A: Thanks for pointing out the typos. Yes, “d_{f}” should be “d_{i}”. We will correct them in the updated version. Here ||d_{i}|| is calculated with the Frobenius norm.
>
> Q5: “In Figure 3, the number of bins used to generate the histogram was not specified as well as whether or not we should count the weights from the batch normalization layer(s).  If given the total number of weights (i.e |𝜭|), this could have given us an insight on how to better generate / replicate the graphs.”
> A: We use 100 bins for all histograms in Figure 3. The histogram does count the weights from BN layers. Since the number of BN parameters are very small in comparison with the total number of weights, it may not significantly change the shape of the histogram the BN weights are not counted. Please refer to our answers to Q2.
>
> Q6: “We were unsure of what the x-axis is representing in figure 4, as it isn’t labelled. We assumed it represented alpha, the parameter used to scale the random high-dimensional gaussian vector , and a vector approach to generate other graphs with which we evaluate the accuracy. Similarly, we assumed the same for Figure 5 on both the x and y axes.”
> A: Yes, the axis of Figure 4 is alpha. The x and y axes of Figure 5 are alpha and beta, which are the step sizes for the two random directions.
>
> Q7: “We didn’t attempt to re-implement “Wide” ResNets as they require fine tuning on the ImageNet dataset which we were unable to train on due to space and computing constraints. ”
> A: The “Wide” ResNets are originally designed for ImageNet but we trained them from scratch on CIFAR-10.
>
> Q8: “It is not specified whether training or testing losses were used to generate the graphs in figure 5.”
> A: All the contours are training losses, it would be interesting to draw test contours.  However, the loss surface being optimized by SGD is the training loss, not the test loss, and so this is what we visualized.
>
> Q9: “For Figures  2,3,4,5,6,7,8, the step-sizes for the alpha and beta values of the different gaussian vectors to generate the mesh grid is not specified.”
> A: We will add the details of the resolutions used in each contours. The default resolutions used for the 2D contours in Figure 5 and 6 is 51x51. We use higher resolutions (251x251) for the ResNet-56-noshort used in Figure 1 to show more details. The resolution for Figure 4 is 401.

---

### Decision · Program_Chairs · 2018-01-29
**ICLR 2018 Conference Acceptance Decision**

**Decision:**

Invite to Workshop Track

**Comment:**

This work proposes an improved visualisation techniques for ReLU networks that compensates for filter scale symmetries/invariances, thus allowing a more meaningful comparison of low-dimensional projected optimization landscapes between different network architectures.

- the visualisation techniques are a small variation over previous works
+ extensive experiments provide nice visualisations and yield a clearer visual picture of some properties of the optimization landscape of various architectural variants

A promising research direction, which could be further improved by providing more extensive and convincing support for the significance of its contribution in comparison to prior techniques, and to its empirically derived observations, findings and claims.